# On the forbidden graphene's ZO (out-of-plane optic) phononic band-analog vibrational modes in fullerenes

Jesús N. Pedroza-Montero [1], Ignacio L. Garzón [2] & Huziel E. Sauceda [3,4 ✉]

The study of nanostructures' vibrational properties is at the core of nanoscience research. They are known to represent a fingerprint of the system as well as to hint the underlying nature of chemical bonds. In this work, we focus on addressing how the vibrational density of states (VDOS) of the carbon fullerene family ($C_n$: $n = 20 \rightarrow 720$ atoms) evolves from the molecular to the bulk material (graphene) behavior using density functional theory. We find that the fullerene's VDOS smoothly converges to the graphene characteristic line-shape, with the only noticeable discrepancy in the frequency range of the out-of-plane optic (ZO) phonon band. From a comparison of both systems we obtain as main results that: (1) The pentagonal faces in the fullerenes impede the existence of the analog of the high frequency graphene's ZO phonons, (2) which in the context of phonons could be interpreted as a compression (by 43%) of the ZO phonon band by decreasing its maximum allowed radial-optic vibration frequency. And 3) as a result, the deviation of fullerene's VDOS relative to graphene may hold important thermodynamical implications, such as larger heat capacities compared to graphene at room-temperature. These results provide insights that can be extrapolated to other nanostructures containing pentagonal rings or pentagonal defects.

[1] Programa de Doctorado en Nanociencias y Nanotecnologías, CINVESTAV, CDMX, México. [2] Instituto de Física, Universidad Nacional Autónoma de México, CDMX, Mexico. [3] Fritz-Haber-Institut der Max-Planck-Gesellschaft, D-14195 Berlin, Germany. [4] Machine Learning Group, Technische Universität Berlin, 10587 Berlin, Germany. ✉email: sauceda@tu-berlin.de

Since the discovery of the $C_{60}$ fullerene in 1985[1], a cascade of theoretical and experimental studies on the physics and chemistry of novel carbon nanostructures emerged. As a consequence of the very interesting properties shown by carbon-based nanomaterials, such as mechanical, electronic, optical, and chemical ones, a great number of fundamental investigations and technological applications have been developed since[2–7]. Despite the plethora of research done on carbon nanomaterials like fullerenes, nanotubes, nanoflakes, etc.[8–16], still, some physicochemical properties have not been fully investigated. In particular, insufficient attention has been paid to the vibrational properties of fullerenes, and how they depend on their size evolution and morphology.

During the last couple of decades, the vibrational properties of metal nanostructures have gained a lot of attention[17–19]. This is not only due to their relevance in the design of nanodevices[20] but also because of a notorious development of vibrational spectroscopies, opening the opportunity to directly compare experimental measurements and theoretical calculations[21]. For example, exploiting the symmetry breaking in supported metal nanoparticles (NPs), Carles' group developed a technique to directly measure the vibrational density of states (VDOS)[22,23]. Such a breakthrough[24] lead to the understanding of intricate experimental results using well-established theoretical results[21,25–29], which link the VDOS as an NP vibrational fingerprint that correlates to the morphology of the system[18,21]. Contrasting these results, organic macro-molecules spectroscopy relies only on Raman and IR spectroscopy to experimentally study them. Hence, given the nature of these experimental measurements, we have access to only a selection of vibrational modes and not to their full VDOS.

A fundamental question in nanoscience is how the transition from molecular (physical and chemical) properties evolve to bulk behavior as a function of the system size. In this regard, several results have been published such as the birth of the localized surface plasmon resonance in Au nanoclusters[30], the evolution of thermodynamical properties on metallic nanoparticles[25,31–33], the formation of surface plasmons in sodium nanoclusters[34], as well as the convergence of molecular vibrational spectra to a bulk-like phonon density of states in metal NPs[35]. In the particular case of metal NPs, a smooth transition of the VDOS from the nanoscale to the bulk has been reported, the study that is missing in the case of carbon nanostructures.

In this work, we present a theoretical investigation of the vibrational properties of fullerenes and their dependence on size ($20 \rightarrow 720$ atoms) at the density functional theory (DFT) theory level. In particular, we analyze the evolution of the fullerene VDOS toward the bulk material (graphene) regime. Analysis that leads to an interesting and counterintuitive difference between the vibrational modes in fullerenes and the phonon bands in graphene: a severe restriction on the out-of-plane (radial) optical band (ZO-modes) in fullerenes in the neighborhood of the $\Gamma$ symmetric point. These results show that the presence of the pentagonal faces in the fullerene family hinders the smooth convergence of their VDOS to the bulk graphene one. Furthermore, these findings hint that pentagonal impurities in graphene can have far more important implications on its vibrational properties than expected.

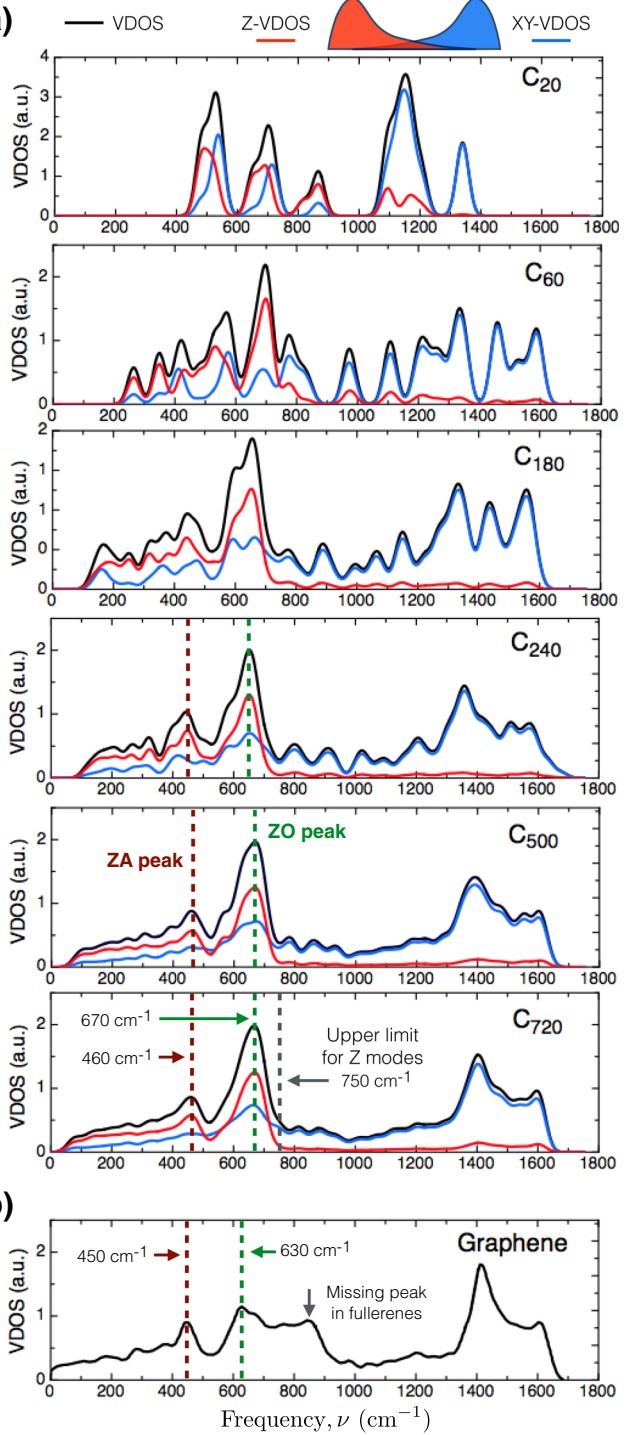

**Fig. 1 Size convergence of the VDOS for the fullerene family. a** Fullerenes' VDOS (black/enclosing curve) as a function of the diameter. The red and blue curves represent the radial and tangential contributions to the total VDOS, respectively. The VDOS was constructed using a Gaussian broadening with a width of 20 cm⁻¹. **b** The bulk graphene VDOS was taken from ref. [65]. The ZA and ZO VDOS peaks are highlighted by dashed lines, as well as other relevant features in the VDOS.

## Results

### Vibrational density of states

*The $C_{20}$ case.* Figure 1a shows the VDOS of fullerenes from 20 to 720 atoms (diameters from 0.4 to 2.6 nm) with $I_h$ symmetry. As a consequence of the finite size of the fullerenes, the frequency spectrum is discrete, and it has a finite *acoustic gap* $\nu_{AG}$ (the

lowest frequency value). The vibrational spectrum extends from $\nu_{AG}$ up to ~1600 cm⁻¹, although in the case of $C_{20}$, it is notorious a smaller frequency distribution range which goes from ~480 cm⁻¹ to 1340 cm⁻¹. It is well known that the $C_{20}$ fullerene is a controversial molecule, in which the reported calculated structure

strongly depends on the utilized level of theory due to its high electron correlation and multiconfiguration character[36–42]. Often, different methodologies differ in the molecular point group of the cage ground state of $C_{20}$, which is reflected on the values of the interatomic forces and consequently in its vibrational spectrum. In particular, the main differences are located near the lowest frequency region. In previous theoretical work, Saito and Miyamoto[36] reported a frequency distribution range that goes from ~115 to ~1435 cm$^{-1}$ using DFT with the hybrid functional B3PW91. Another study by Schütt et al. recently reported a vibrational spectrum ranging from ~100 to ~1400 cm$^{-1}$ using a machine learning method based on deep neural networks, SchNet[41,42], trained on DFT(PBE + vdW$^{TS}$)[43] level of theory. Similar results can be obtained with other machine learning approaches[44–46]. In these two works, the $\nu_{AG}$ is lower than in our case because the lower symmetry group considered as a global minimum, $D_{3h}$ instead of $I_h$. In the case of the highest frequency, also known as cut-off frequency $\nu_{COF}$, the lower value displayed in $C_{20}$ compared to other fullerenes is attributed to the nature of the carbon–carbon bond in $C_{20}$ coming from the fact that the geometrical structure is only formed by pentagonal faces. The rest of the studied fullerenes are more electronically stable and different levels of theories give consistent results, showing a smoother transition between different sizes.

*The hypothesis.* Based on the fact that fullerenes are single-layer carbon cage structures, it is natural to assume that in the limit of large fullerene diameters, the VDOS should converge to the graphene's VDOS given its single layer character. To explore this hypothesis, in Fig. 1a we show the VDOS evolution with the size of the fullerenes and its comparison to bulk graphene (Fig. 1b). Here, we can think of the graphene case as the case of fullerene with diameter → ∞, where the twelve pentagonal faces are completely diluted. From intermediate sizes, such as the $C_{240}$, we can see that the line shape of the VDOS already has the main features of the biggest fullerene in our study (i.e., $C_{720}$). Furthermore, the fullerenes display the higher intensity (main) peak in the VDOS around 670 cm$^{-1}$, a characteristic feature that emerges from the 60 atom structure.

*Types of vibrational modes.* When comparing the VDOSs of the fullerene family and graphene, it is observed that the $C_{720}$ fullerene presents most of the features displayed by bulk graphene, with the obvious difference of the finite acoustic gap. However, it is worth highlighting the mismatch located in the range of ~540–910 cm$^{-1}$, where the graphene peak around ~850 cm$^{-1}$ is missing in the fullerene family (See Figs. 1b and 2). In order to understand this difference, first, we analyze the VDOS of graphene. Graphene is a 2D material that presents vibrations/phonons that can be classified into two main groups, in-plane (XY) and out-of-plane (Z) vibrations (See Fig. 3). This means that in XY phonons the atomic displacement will be contained in the

same plane of the graphene, whereas in Z phonons the displacements will be perpendicular to the graphene plane. Such a decomposition of the VDOS for graphene has been discussed by Paulatto et al.[47], indicating that the Z vibrational modes can be divided into acoustic (ZA) and optic (ZO) vibrations, and that their respective frequency intervals are [0, ~540] and [~540, ~910] cm$^{-1}$[47–49]. These two bands are highlighted in Fig. 3a, ZA in red and ZO in green, as well as their own density of states, VDOS$^{ZA}$ and VDOS$^{ZO}$, respectively (See Fig. 3b). It is worth stressing that all Z vibrations/phonons are confined to frequencies *below* 910 cm$^{-1}$.

By using this idea in the case of fullerenes, we can separate their VDOS contributions from normal modes with radial atomic displacements or out-of-shell (Z, red lines in Fig. 1a) from those containing displacements in-shell (XY, blue lines in Fig. 1a). In the case of the XY-VDOS for fullerenes (blue line), we observe a smooth line shape spreading along the whole frequency spectrum since it contains all the in-shell vibrational modes: TA, LA, TO, and LO. On the other hand, for the Z modes (red line) shown in Fig. 1a from top to bottom, we can see that the Z-VDOS converge to a smooth flat curve above 800 cm$^{-1}$, displaying two well-defined peaks at 460 and 670 cm$^{-1}$, consistent with those observed in graphene (See Fig. 1b).

**The forbidden wavenumbers in the ZO branch in fullerenes.** As stated in the previous section, the Z modes in fullerenes have a cut-off frequency of ~750 cm$^{-1}$, while the corresponding value for graphene is ~910 cm$^{-1}$ (see Fig. 3). Figure 2 shows the overlap between the VDOS of $C_{720}$ and graphene, clearly showing the fullerene's missing region in red. By analyzing the phonon bands of graphene, we hypothesize that the missing peak is due to the lack of modes that would come from the highest frequency region of the ZO branch (centered in the Γ symmetry point and marked by a red square in Fig. 3a). The origin of this is the fact that some symmetries of the wave vectors (i.e., vibrational eigenvectors) are not allowed in fullerenes. To illustrate the kind of vibrational modes in the ZO (out-of-plane optical) branch, in Fig. 4a we show the highest frequency mode in the branch which corresponds to the high-symmetry Γ point. This consists of anti-phase out-of-plane atomic displacements between first neighbors. Such vibrational mode is only allowed because of the hexagonal lattice of graphene (i.e., it has an even number of atoms in the ring), while in the case of fullerene family the presence of the pentagons prohibits the existence of this kind of vibrational mode. A schematic example is constructed in Fig. 4b, where the mode in Fig. 4a is being constructed from top to bottom in a lattice with a pentagonal ring, which then inevitably ends up in two neighboring atoms moving in phase. Therefore, *the incapability of lattices with pentagons to host certain types of ZO modes limits the value of the cut-off frequency of Z-modes $\nu_Z^{\max}$ in fullerenes.* In fact, the frequency value $\nu_Z^{\max}$ of the vibrational mode and its shape (i.e., its eigenvector) not only depends on the size but also on the molecular symmetry point group (see Supplementary Fig. 1). In general, the 12 pentagonal faces in fullerenes severely restrict high-frequency Z optical modes but allows the creation of slower optical modes (hence the amplification of the ZO peak in Figs. 1 and 3b).

This should also be true for bulk material with hexagonal lattice with pentagonal defects since the odd number of atoms in the pentagonal ring hinders such highly symmetric vibrational motion, originating a lower frequency in-phase motion between first neighbors (Fig. 4b, in green).

Figure 3 highlights the contributions of the ZO (green) and ZA (red) phonon bands to the total VDOS, and their direct comparison to the fullerenes' Z-VDOS is presented. From this

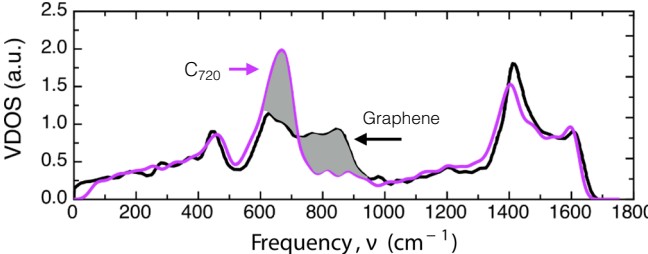

**Fig. 2 VDOS comparison between fullerene $C_{720}$ and graphene.** Bulk graphene VDOS was taken from ref. [65]. The suppressed phonon ZO branch in fullerenes is highlighted by the gray area.

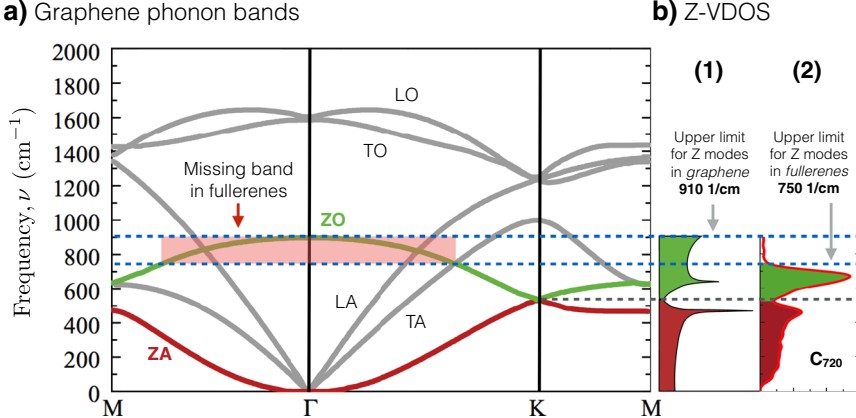

**Fig. 3 Comparison of Z-phonon-bands between bulk graphene and fullerenes. a** Graphene phonon bands and their mode classification. The out-of-plane (Z) phonons are highlighted in red for the acoustic band (ZA) and in green for the optical (ZO) band. **b**-1 ZO-band and ZA-band contribution to the graphene total VDOS. **b**-2 Radial (Z)-VDOS in $C_{720}$ fullerene. Graphene's VDOS was taken from ref. [47]. The missing frequency region of the ZO-bulk-analog phonons missing in the fullerene family is shadowed in **a** and highlighted by dashed lines.

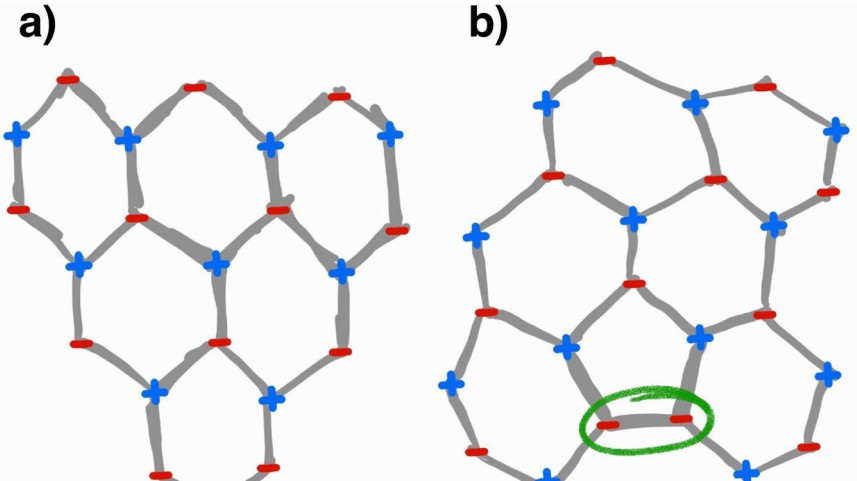

**Fig. 4 Pictorial representation of the highest frequency out-of-plane oscillation. a** The Highest frequency ZO vibrational mode[49] in graphene. **b** Construction of a ZO vibrational mode in a hexagonal lattice with a pentagonal defect. In **b**, we can see that the presence of pentagons will produce an in-phase motion between first neighbors (circled in green), and consequently, the highest frequency ZO vibrational mode for fullerenes will be lower than the one in graphene. The blue plus and red negative signs represent atomic motions going in and out of the plane, respectively.

figure, we can clearly see the region of the ZO phonon branch that would correspond to the missing vibrational modes in the fullerene family.

**Visualization of highest Z modes**. In order to visually understand what has explained above, it is interesting to analyze how the ZO modes look like in fullerenes, in particular, the one corresponding to the maximum allowed frequency, $\nu_Z^{max}$. The family of these modes is displayed in Fig. 5a. From here, we can see that starting from $C_{180}$ to $C_{720}$, the shape of the associated eigenvector $\mathbf{v}_Z^{max}$ follows similar patterns. The shape of the vibrational modes can be contrasted to the highest frequency ZO phonon (Γ point) in Fig. 4a. Interestingly, the $\nu_Z^{max}$ value converges to ~710 cm$^{-1}$ for all fullerenes in our study, except for $C_{20}$ given the fundamental differences in the structure and chemical bonds. Figure 5b shows the range of frequencies in which the ZO branch is defined in graphene (highlighted in green in Fig. 3) and where the $\nu_Z^{max}$ values lay relative to this reference interval. It is expected that as the diameter of the fullerene grows, the $\nu_Z^{max}$ will slowly start increasing towards the graphene ZO branch maximum value

given that the contribution of the area of the 12 pentagons relative to the rest of the fullerene surface will start diluting.

**Thermodynamic implications**. In order to show that the vibrational peculiarities of fullerenes' VDOS have definitive thermodynamical implications, in this section, we present a direct comparison between $C_{720}$'s and graphene's specific heat. Hence, we will focus on the comparison made in Fig. 2.

Before going to our particular case, we should summarize some equivalent results from metallic NPs. It is a well-established result that metal NP has a larger heat capacity compared to their bulk counterpart for a short interval at low temperatures, e.g., ~3–40 K in the case of Au NP[35]. This is due to the higher population of normal modes in the lower frequency part of the VDOS compared to the bulk material, meaning that the VDOS for metal NP grows as $g(\nu) \sim \nu^n$ with $n > 2$ at the lower part of the spectrum. Now, in the case of fullerenes, we can see different behavior from Fig. 2 compared to metal NPs, where essentially the $C_{720}$'s VDOS recovers most of the features from the graphene case with the difference of a late start of the VDOS due to the acoustic gap and the already discussed compressed ZO band peak

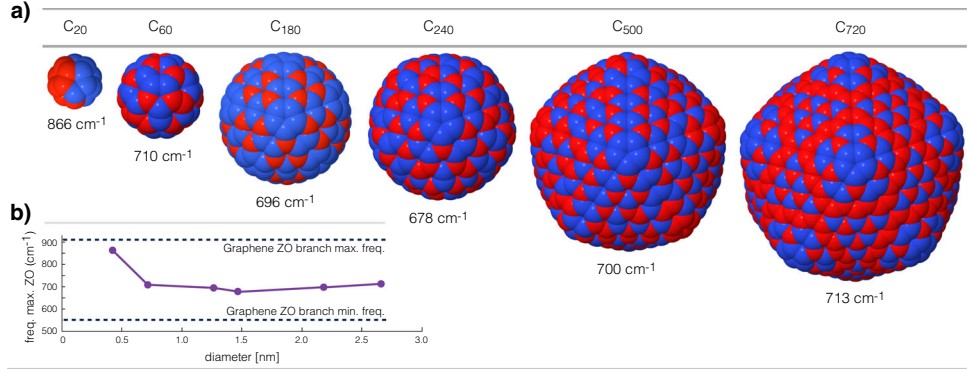

**Fig. 5 Shape of the highest frequency ZO vibrational modes in the fullerene family. a** Visualization of the highest frequency purely-radial vibrational modes, which correspond to the phonons in the ZO branch in graphene. The red and blue colors indicate phase and anti-phase radial atomic displacements, respectively. **b** Dependence of the corresponding vibrational frequencies versus the fullerenes' diameter. The dashed lines indicate the frequency limits of the ZO branch in graphene.

around 670 cm$^{-1}$. An analysis of Fig. 2 suggests that the potential thermodynamical implications of the compressed ZO band in fullerenes will come at higher temperatures which is an unusual characteristic compared to other systems reported in the literature[25,26,35]. To quantify such effects, we analyze the thermodynamics of both systems, C$_{720}$, and graphene, to elucidate their relative behavior.

In general, thermodynamical variables are obtained by a cumulative process of the vibrational properties of the system, meaning that computing such quantities involve a sum or an integral over the vibrational spectrum weighted by a function. As an example, let's consider the specific heat,

$$C(T)/k_{\mathrm{B}} = \int d\nu' f(\nu', T) g(\nu') \tag{1}$$

where $g(\nu)$ is the VDOS function, $k_{\mathrm{B}}$ is the Boltzmann's constant, and the weight function $f(\nu, T)$ comes from the temperature derivative of the Bose–Einstein distribution function,

$$f(\nu, T) = \left(\frac{h\nu}{2k_{\mathrm{B}}T}\right)^2 \sinh^{-2}\left(\frac{h\nu}{2k_{\mathrm{B}}T}\right). \tag{2}$$

Now, by using Eq. (1) we can quantify the thermodynamic deviations of the fullerene relative to graphene by using,

$$\frac{\Delta C(T)}{k_{\mathrm{B}}} = \int d\nu' f(\nu', T)(g_{\mathrm{fuller}}(\nu') - g_{\mathrm{graphene}}(\nu')), \tag{3}$$

as shown in Fig. 6. Then, as a reference, we compare these results to the pure cumulative function of the difference of the two vibrational spectra,

$$\Delta g(\nu) = \int_0^\nu d\nu'(g_{\mathrm{fuller}}(\nu') - g_{\mathrm{graphene}}(\nu')), \tag{4}$$

displayed in blue in Fig. 6. From these results, we can see a clear correlation between the two functions, which then suggests that the different rates of accumulation are at the origin of the distinct behavior in the heat capacities. To better understand the results, we have highlighted three relevant temperature values in this figure: (1) marks that at ≈54 K, graphene's specific heat has its maximum gain compared to the fullerene. This is originated from the acoustic gap in finite systems (see Fig. 2), meaning that the accumulation process starts earlier for graphene. (2) Then, at ≈170 K the rapid growth of the ZO peak in the fullerene equals the cumulative value in graphene, and from that temperature onward the fullerene heat capacity will be larger than the graphene value, reaching its maximum at (3) ≈362 K.

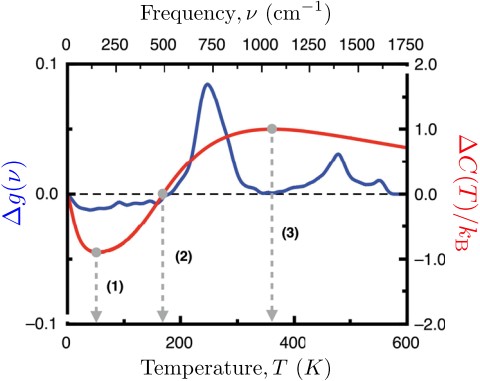

**Fig. 6 Thermodynamical implications of the pentagonal facets in the fullerene family.** Temperature-dependent difference of the heat capacities (red curve) between the C$_{720}$ fullerene and graphene, Eq. (3), scaled by 10$^3$ (the right side is its corresponding y-axis). The blue curve corresponds to the cumulative value of the difference between their VDOSs as a function of the frequency, Eq. (4) (the left side is its corresponding y-axis). The dashed vertical lines indicate relevant temperature values: (1) 54 K, graphene's heat capacity reaches its maximum value relative to the C$_{720}$'s one; (2) 170 K, equality of heat capacities; (3) 362 K, C$_{720}$'s heat capacity reaches its maximum value relative to graphene's one.

These results demonstrate that indeed the excess of vibrational modes created in the ZO band by the pentagonal faces in fullerenes (see Fig. 3) have an impact on the system's thermodynamics and even overcome the disadvantage imposed by the acoustic gap. Furthermore, it is worth noting that contrary to other systems such as metal NPs, fullerenes high density of vibrational modes at the ZO band region plays a strategic role in providing them the unique property of having larger heat capacities than the bulk material at temperatures comparable to room temperature and higher. Characteristic that, to the best of our knowledge, is the first reported system having such property.

## Discussion

The present analysis, on the evolution of fullerenes' VDOS with the diameter and its similarities and deviations from the bulk graphene VDOS, reveals an interesting geometrical constraint, on the type of vibrational modes that fullerenes can host. Unlike the size evolution towards the bulk behavior obtained for the VDOS of metal nanoparticles[21], in the present case, there will not be a full convergence to the graphene VDOS since the pentagonal facets are required to close the cage structures of fullerenes. In

particular, in terms of the phononic bands in graphene, the ZO-equivalent normal mode oscillations are greatly affected by the presence of pentagonal facets, which sets a lower frequency ($\sim$750 cm$^{-1}$) and dynamically different ZO cut-off eigenstate (comparison between Figs. 4a and 5), as well as a narrower ZO band ($\sim$43%). This compression of the ZO-band in fullerenes, reshapes the VDOS, as shown in Fig. 2, which means that necessarily this will have implications on the thermodynamics of the system. In particular, we have shown evidence of such consequences for the heat capacity (see Fig. 6), where it was found that at low temperatures, graphene has a higher heat capacity compared to fullerenes given their finite size. Nevertheless, from temperatures above 170 K, the $C_{720}$ fullerene's heat capacity surpasses the reference graphene value due to the cumulative boost induced by the compressed ZO-band in fullerenes. Hence, rendering fullerenes as the first systems reported to have larger heat capacities than its bulk material counterpart. On a more general note, this type of effect is to be expected in other thermodynamical quantities given that many of them are defined by cumulative analytical formulas, hence providing great relevance to the features characterizing the vibrational properties of fullerenes.

Regarding the dependence of the vibrational properties of fullerene with their size, an important assumption in our study was that we can take graphene's VDOS as a fullerene with a very large diameter, where the effects of the pentagonal faces are not relevant. Hence, in such an idealized scenario, this would imply that $\lim_{d\to\infty}\nu_Z^{\max}(d) = \nu_{ZO,\mathrm{Graphene}}^{\max}$, where $d$ is the diameter of the fullerene. Such convergence is very slow as can be appreciated in Fig. 5b. Nevertheless, we have to remember that normal modes are collective atomic oscillations, meaning that either the interatomic interaction in the pentagonal faces strengthen with the size (thereby increasing its frequency) or the pentagonal faces slowly become nodes in the normal mode oscillation i.e., in the $\mathbf{v}_Z^{\max}$ eigenvector, allowing then higher frequency oscillations localized in the hexagons. By analyzing the atomic amplitudes of the normal modes shown in Fig. 5, we have found evidence of the latter option, as reported in Supplementary Table 1. In concrete, the oscillation amplitudes of the atoms in the pentagonal facets of the fullerenes for the $\mathbf{v}_Z^{\max}$ eigenvector reduces as the diameter increases. In an ideal scenario, such a result would imply that, if the vibrational mode corresponding to the Γ symmetric point in the ZO phonon band exists in a graphene material with pentagonal defects, those defects will have null amplitudes. Nevertheless, in a more realistic case, pentagonal defects in monolayer carbon structures come with other geometrical features[50,51] which implies that a thorough analysis is required to see how the results presented here translate to such case, a topic which goes beyond the scope of this article, then will be left as future work.

On the other hand, it is worth mentioning that the presence of pentagonal defects in finite-size carbon nanoflakes and nanotubes also modifies the VDOS of the corresponding pristine nanostructures. For example, our VDOS calculations for a 96-atom carbon nanoflake with and without a pentagonal defect show differences in the corresponding VDOS in the region of 600–800 cm$^{-1}$, where the modes associated with ZO phonons would be present. Similar behavior was obtained for defective and pristine 400-atom nanotubes, indicating that pentagonal defects induced differences in their VDOS in the 450–600 and 1200–1600 cm$^{-1}$ frequency regions. This result is in contrast with a recent theoretical report, based on a semiempirical model for the carbon atom interactions, where the difference in the VDOS was obtained at smaller frequencies[52]. It is also useful to analyze the role of the pentagons in the VDOS of fullerenes and pristine carbon nanotubes of similar size. For example, the comparison between the VDOS of the $C_{500}$ fullerene and a 400-atom pristine nanotube shows different profiles in the 400–1000 cm$^{-1}$ region, that might be attributed to the presence of

pentagons in the fullerene, although the geometric structures also induce variations in the VDOS. Evidence of these results is shown in Supplementary Figs. 1 and 2 for the VDOS calculations for pristine and defective carbon nanoflakes and nanotubes, respectively. In addition, as a reference, Supplementary Fig. 3 displays the comparison of the VDOS for the fullerene 500 and nanotube 400.

To conclude, we have presented here thorough analysis of the peculiarities in the vibrational properties of fullerenes relative to the bulk graphene, and their different thermal behavior due to the presence of a compressed *ZO-band* in fullerenes. This work opens new aspects and peculiarities of carbon nanostructures which then generate additional questions on the physicochemical implications that pentagonal or other types of ring defects would produce in graphene and other carbon systems, and also contributes to gain further insight into the physical properties of carbon nanostructures.

## Methods

**Ab initio calculations**. All computations were done at the density functional theory (DFT) level within the generalized gradient approximation (GGA), using the Perdew-Burke-Ernzerhof (PBE) parameterization[53]. A split valence basis set (def2-SVP) and the pseudopotential for carbon with 4 valence electrons corresponding to ECP2SDF were used[8,54], as implemented in the TURBOMOLE code[55]. To guarantee accurate frequencies' calculation, the structures were optimized with a force tolerance of $10^{-6}$ Hartree Bohr$^{-1}$. The computed harmonic frequencies were validated with reported calculations for carbon dimer's bond length, binding energy, and vibrational frequency[56–62]. In addition, the whole vibrational spectrum of $C_{60}$ fullerene was calculated and compared with experimental results[63] for validation. The VDOS was constructed using a Gaussian broadening with a width of 20 cm$^{-1}$ of the $3N-6$ frequencies.

**Separating the Z-VDOS contribution**. The selection of purely radial vibrational modes in fullerenes and their contribution to the VDOS was estimated by performing a radial projection of the whole vibrational spectrum,
$Z-\mathrm{VDOS}(\nu) = \sum_{i=1}^{3N-6}\langle\mathbf{x}|\mathbf{v}_i\rangle\delta(\nu-\nu_i)$, where $\mathbf{x}$ is a radial vector and $\{\mathbf{v}_i\}$ is the set of eigenvectors. Then, the in-plane (or more precisely for fullerenes, in-shell) contribution is computed via XY-VDOS = VDOS–Z-VDOS.

## Data availability

The TURBOMOLE code outputs containing optimized structures and their frequencies generated for this study have been deposited in the NOMAD repository (https://doi.org/10.17172/NOMAD/2021.06.12-1) (ref. [64]).

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

## Acknowledgements

We thank Dr. Luis E. Gálvez-González for part of the additional calculations. I.L.G. thanks support from DGTIC-UNAM under Project LANCAD-UNAM-DGTIC-049, DGAPA-UNAM under Project IN106021, and CONACYT-Mexico under Project 285821. J.N.P.M. thanks CONACYT-Mexico for an SNI Research Assistant fellowship and the financial support from CONACYT Project No. 177981. H.E.S. also works at the BASLEARN-TU Berlin/BASF Joint Lab for Machine Learning, co-financed by TU Berlin and BASF SE. H.E.S. thanks Prof. Dr. Klaus-Robert Müller for helpful and inspiring discussions.

## Author contributions

H.E.S. conceptualized the project; H.E.S. and I.L.G. designed the research; J.N.P.M. and H.E.S. performed the calculations; All the authors analyzed and interpreted the data and contributed to the writing of the manuscript.

## Funding

## Competing interests

The authors declare no competing interests.
