## [Peer Review File · Communications Chemistry]

Reviewers' comments:

Reviewer #1 (Remarks to the Author):

The authors present the study of different fullerenes' vibration properties ($n=20\sim 720$). They address the vibrational density of states (VDOS) from the molecular to the bulk material (graphene). It is found that the fullerene's VDOS smoothly converges to the graphene characteristic shape-line with the only noticeable discrepancy in the frequency range of the out-of-plane optic (ZO) phonon band in graphene. They also obtain some main results: 1) The pentagonal faces in the fullerenes impede the existence of the analog of the high frequency graphene's ZO phonons, 2) which in the context of phonons this could be interpreted as a compression (by 43%) of the ZO phonon band by decreasing its maximum allowed radial-optic vibration frequency. 3) As a result, the deviation of fullerene's VDOS relative to graphene should result on important thermodynamical implications. In summary, I think this manuscript is well written, and some revisions are needed before recommendations.

1. The authors compared fullerenes with graphene. It is known that fullerenes have pentagonals while pristine graphene has no pentagonals. Additionally, they point out the pentagonals' effects on VDOS. Herein, I think the authors need to investigate some graphene models with pentagonals to compare and confirm the pentagonals' effects.

2. Besides fullerene and graphene, some typical carbon nanotubes' VDOS may also need to extend the discussion.

Reviewer #2 (Remarks to the Author):

This paper reports on a density functional theory study of the vibrational density of states (VDOS) of fullerenes and compares them with that of graphene. It is observed that the VDOS of the fullerenes (from C₂₀ to C₇₂₀) converges to that of graphene, except for a discrepancy in the out-of-plane optical phonon branch. This difference is ascribed to the presence of pentagonal faces of the fullerenes, which are incompatible with the highest frequency out-of-plane optical modes. In my opinion this is an original and interesting study; the methodology is suitable; and the interpretation of the results is plausible. The findings have broader relevance in the context of phonon engineering in nanostructures and 2D-materials. I recommend publication of this manuscript after the authors have considered the following optional recommendations.

The role of the pentagons to in the different VDOS of fullerenes and graphene is well motivated. It is, however, not clear if confinement effects/finite size of fullerene also are a cause of differences, besides from the obvious finite acoustic gap. If the authors also compute the phonon density of carbon nanotubes at the same level of theory, this could allow to differentiate contributions from pentagons and confinement. In this context, the authors may want to refer to <https://doi.org/10.1016/j.carbon.2017.03.030>

As the authors discuss, one expects that the fullerene Z modes will evolve to that of graphene if the diameter of the fullerene grows and the pentagons become diluted. Convergence seems to be slow (inset figure 5). Possibly, the authors can quantify the slow convergence, for instance by comparing it with the fraction of in phase nearest neighbor radial atomic displacements in the highest frequency vibrational modes (graphs in figure 5), which is zero in graphene (figure 4 a). Also the initial decrease of the maximal frequency ZO mode from $d = 0.5$ nm to 1.5 nm (inset figure 5) must have another origin and is not discussed.

Some language polishing of the manuscript is required. Some sentences are badly formulated such as the last sentence of the abstract, the last sentence of the first paragraph, and last sentence of the conclusion (before methodology).

Reviewer #3 (Remarks to the Author):

I do not recommend the manuscript "On the forbidden graphene's ZO (out-of-plane optic) phononic band-analog vibrational modes in fullerenes" by Saucedo and collaborators for publication in Communications Chemistry.

The paper is technically sound and the manuscript is clearly written and well organized.

However, I find the scope of the results and discussion too limited for publication in a high-impact journal such as CommsChem.

There is really only one novel result in this work: the absence of certain phonon modes that are present in a pristine graphene sheet, and that do not occur in fullerenes ranging from C₂₀ to C₇₂₀, because of the presence of the pentagons. Pentagons lead to a frustration of the pattern of displacements of the corresponding radial vibrational modes in fullerenes.

The authors no more than allude to "important thermodynamical implications" of the absence of these modes, without explaining what would such implications be. Furthermore, they speculate that in the limit of graphene as a fullerene of infinite radius with very diluted pentagons, the absent modes would emerge with the pentagons having zero-amplitude displacements.

These speculations are inconsistent with the fact that an isolated pentagon in a graphene sheet is a disclination, i.e., it induces topological conical curvatures, and even in a sheet with null net curvature (with the same number of isolated disclinations and anti-disclinations) local curvatures would be associated with the isolated pentagons. The scenario of an isolated pentagon having zero-amplitude in the said phonon modes is very unlikely.

Moreover, it seems to me that all that it takes is for an odd number of vertices of the pentagons to have zero-amplitude for the mode to be present in a graphene sheet with isolated diluted pentagons, at least from the point of view of the frustration of the pattern of displacements of the modes in question.

To summarize, I do not find the scope of the results and the discussion to warrant publication in a high-impact journal such as Communications Chemistry.

Response to reviewers' comments and suggestions

Reviewer #1 (Remarks to the Author):

The authors present the study of different fullerenes' vibration properties ($n=20\sim 720$). They address the vibrational density of states (VDOS) from the molecular to the bulk material (graphene). It is found that the fullerene's VDOS smoothly converges to the graphene characteristic shape-line with the only noticeable discrepancy in the frequency range of the out-of-plane optic (ZO) phonon band in graphene. They also obtain some main results: 1)The pentagonal faces in the fullerenes impede the existence of the analog of the high frequency graphene's ZO phonons, 2)which in the context of phonons this could be interpreted as a compression (by 43%) of the ZO phonon band by decreasing its maximum allowed radial-optic vibration frequency. 3)As a result, the deviation of fullerene's VDOS relative to graphene should result on important thermodynamical implications. In summary, I think this manuscript is well written, and some revisions are needed before recommendations.

We thank Reviewer 1 for her/his positive comments and suggestions. Here, we respond to all the specific actions requested.

1. The authors compared fullerenes with graphene. It is known that fullerenes have pentagonals while pristine graphene has no pentagonals. Additionally, they point out the pentagonals' effects on VDOS. Herein, I think the authors need to investigate some graphene models with pentagonals to compare and confirm the pentagonals' effects.

Following the Reviewer 1 suggestion, we have calculated the VDOS for two graphene nanoflakes with and without pentagonal defects. The optimized geometries, as well as their corresponding VDOS are shown in the figure below. The difference between the red and blue curves clearly displays the effect of the two pentagonal defects, inserted in the defective nanoflake. It should be noted that the main differences are in the region of $600\text{-}800\text{ cm}^{-1}$, where the modes associated with the ZO phonons would be present. Also, a higher intensity in the VDOS of the pristine nanoflake is obtained around 1420 cm^{-1} . These results show that the existence of pentagons in graphene models indeed modifies the VDOS of its hexagonal lattice. A new paragraph describing these results was inserted in the revised manuscript together with Figure R1 included in the SI section.

Figure R1. Top panel: Left: Optimized structure of a graphene pristine nanoflake. Right: Optimized structure of a graphene defective nanoflake. In the latter case, two pentagonal-heptagonal defects were inserted at the center of a pristine nanoflake with 96 carbon atoms. Hydrogen atoms were used to saturate the surface dangling bonds. Bottom panel: VDOS of the pristine (blue curve) and defective (red curve) carbon nanoflakes.

2. Besides fullerene and graphene, some typical carbon nanotubes' VDOS may also need to extend the discussion.

We have also calculated the VDOS of a 400-atom carbon nanotube with and without pentagonal-heptagonal defects. Figure R2 shows the optimized geometries of both a pristine and a defective carbon nanotube C400, and their corresponding VDOS. In these cases, it is also evident that the presence of two pentagonal-heptagonal defects induced differences in the VDOS in the 450-600 and 1200-1600 cm^{-1} regions. In the low-frequency region, it was found modes corresponding to stretching motions of the atoms neighboring the pentagons, whereas for the higher frequencies, rotating modes including the pentagonal and heptagonal faces are present. These results together with Figure R2 were included in the SI section of the revised manuscript.

On the other hand, according to the work published by A. J. Pool et al. Carbon 118, 58 (2017), the presence of pentagonal defects in pristine nanotubes modify the VDOS, particularly in the low-frequency L and L' modes region. This new information and the corresponding reference was inserted in the revised manuscript.

Figure R2. Top panel: Left: Optimized structure of a pristine nanotube. Right: Optimized structure of a defective nanotube. In the latter case, two pentagonal-heptagonal defects were inserted at the surface of a pristine nanotube with 400 carbon atoms. Bottom panel: VDOS for the pristine and defective C400 nanotube.

Reviewer #2 (Remarks to the Author):

This paper reports on a density functional theory study of the vibrational density of states (VDOS) of fullerenes and compares them with that of graphene. It is observed that the VDOS of the fullerenes (from C20 to C720) converge to that of graphene, except for a discrepancy in the out-of-plane optical phonon branch. This difference is ascribed to the presence of pentagonal faces of the fullerenes, which are incompatible with the highest frequency out-of-plane optical modes. In my opinion this is an original and interesting study; the methodology is suitable; and the interpretation of the results is plausible. The findings have broader relevance in the context of phonon engineering in nanostructures and 2D-materials. I recommend publication of this manuscript after the authors have considered the following optional recommendations.

We thank Reviewer 2 for her/his positive comments on our manuscript. Below, we address all the mentioned comments and suggestions.

The role of the pentagons in the different VDOS of fullerenes and graphene is well motivated. It is, however, not clear if confinement effects/finite size of fullerene also are a cause of differences, aside from the obvious finite acoustic gap. If the authors also compute the phonon density of carbon nanotubes at the same level of theory, this could allow to differentiate contributions from pentagons and confinement. In this context, the authors may want to refer to <https://doi.org/10.1016/j.carbon.2017.03.030>

The reference mentioned by Reviewer 2 has been included and discussed in the revised manuscript.

In order to show the confinement/finite size effects of fullerenes mentioned by Reviewer 2, Figure 2 of the original manuscript displays the size dependence of the VDOS for fullerenes with 20-720 carbon atoms. All fullerene structures contain 12 pentagons despite their increasing size. In Figure 2, it can be appreciated the evolution towards bulk behavior (graphene) and the finite size effects.

As requested by Reviewer 2, we also show below, in Figure R3, the comparison between the calculated VDOS for the C500 fullerene and a pristine (without pentagonal defects) nanotube C400. This comparison indicates that the main differences in the VDOS profiles appear in the 400-1000 cm^{-1} region. In particular, the intense peak at 680 cm^{-1} for the fullerene C500 decreases in intensity for the C400 nanotube. It is concluded that the main reason for these differences in the VDOS might be attributed to the presence of pentagons in the fullerene, although the geometric structure also would induce other variations in the VDOS profiles. The vibrations in the C400 nanotube with frequencies around 700 cm^{-1} correspond to highly symmetric out-of-plane and twisting modes. These results and Figure R3 were included in the SI section.

Figure R3. Calculated VDOS for (top panel) a C400 nanotube and (bottom panel) C500 fullerene.

As the authors discuss, one expects that the fullerene Z modes will evolve to that of graphene if the diameter of the fullerene grows and the pentagons become diluted. Convergence seems to be slow (inset figure 5). Possibly, the authors can quantify the slow convergence, for instance by comparing it with the fraction of in phase nearest neighbour radial atomic displacements in the highest frequency vibrational modes (graphs in figure 5), which is zero in graphene (figure 4 a).

As mentioned by Reviewer 2, indeed, the convergence of the vibrational frequency for the max Z mode is quite slow and because of that it is complicated just from the inset in Figure 5 to draw more information. Hence, we thank Reviewer 2 for the nice suggestion. Then, based on the Reviewer's request, we have computed a measure that correlates the size of the fullerene (number of atoms) with the evolution of the shape of the max freq. Z mode eigenvector: $\langle \text{Atomic_amplitude_in_pentagon} \rangle / \text{Max_Amplitude}$. Here, we have taken the mean of the atomic displacement amplitudes at the pentagonal faces and divided it by the maximum atomic oscillation amplitude in the eigenvector to compute their ratio. This is shown in Table S1. This shows that indeed the relative amplitude of the atomic oscillations around the pentagonal faces decrease in a global manner as the fullerene size increases. Hence, this measure further supports our findings. Table S1 and its description has been added to the Supporting Information section.

Table S1.

# atoms	$\langle \text{Ampl_pentag_face} \rangle / \text{Max_Ampl}$	frequency[1/cm]
60	0.515	710.63
180	0.207	696.46
240	0.373	678.44
260	0.102	684.62
500	0.069	699.71
720	0.096	713.04

Also the initial decrease of the maximal frequency ZO mode from $d = 0.5$ nm to 1.5 nm (inset figure 5) must have another origin and is not discussed.

Indeed, the underlying nature of the chemical bonds in the C20 fullerene is quite different from the rest of the fullerene sizes, this is the main reason why it shows a different behaviour. In fact, this system is considered as highly correlated (i.e. it has a large contribution for electronic correlation). In section II-A-1 of the original manuscript, we described in more detail this system.

Some language polishing of the manuscript is required. Some sentences are badly formulated such as the last sentence of the abstract, the last sentence of the first paragraph, and last sentence of the conclusion (before methodology).

As requested by Reviewer 2, the English style of the manuscript was revised, paying special attention to the mentioned paragraphs.

Reviewer #3 (Remarks to the Author):

I do not recommend the manuscript "On the forbidden graphene's ZO (out-of-plane optic) phononic band-analog vibrational modes in fullerenes" by Saucedo and collaborators for publication in Communications Chemistry.

The paper is technically sound and the manuscript is clearly written and well organized.

However, I find the scope of the results and discussion too limited for publication in a high-impact journal such as CommsChem.

There is really only one novel result in this work: the absence of certain phonon modes that are present in a pristine graphene sheet, and that do not occur in fullerenes ranging from C₂₀ to C₇₂₀, because of the presence of the pentagons. Pentagons lead to a frustration of the pattern of displacements of the corresponding radial vibrational modes in fullerenes.

We thank Reviewer 3 for her/his helpful comments on our manuscript. We have taken into account all the comments and criticisms in order to generate a revised manuscript with additional calculations and results. We expect that the revised (enhanced) manuscript will satisfy the requirements mentioned to deserve publication in CommsChem.

The authors no more than allude to "important thermodynamical implications" of the absence of these modes, without explaining what would such implications be.

Reviewer 3 is right, since no thermodynamic results were reported in the original manuscript and we thank her/him for this comment. In the revised manuscript, we have added results from new calculations on the temperature dependence of the heat capacity of fullerenes, and its comparison with that one calculated for graphene. These new calculations were done within the harmonic approximation using the calculated VDOS for such systems.

Figure R4 (top panels) displays the heat capacity (red curves) and VDOS (blue curves) of the C720 fullerene (left) and graphene (right), respectively. In order to detect more clearly a distinct behaviour in the heat capacity, the bottom panel shows the difference of such quantities (red curve). The blue curve in the bottom panel displays the difference in the integral of VDOS as a function of frequency, between the C720 fullerene and graphene, which is the origin of the distinct behaviour in the heat capacities. In fact, from Figure R4 (bottom panel), it is evident that the graphene heat capacity is larger than that of the C720 fullerene up to ~170 K, but at higher temperatures it was obtained an opposite result (see red curve). This distinct behaviour is related to the difference in the integrated or cumulative VDOS profiles (blue curve), that is mainly due to the compressed ZO band in the VDOS of the C720 fullerene (note, for example, the peak centered at 750 1/cm in the blue curve in the bottom panel).

These results demonstrate that indeed the excess of vibrational modes created in the ZO band by the pentagonal faces in fullerenes have an impact on the system's thermodynamics and even overcome the disadvantage imposed by the acoustic gap.

Furthermore, it is worth noting that contrary to other systems comparing finite particles/clusters vs bulk material, such as metal nanoparticles, fullerenes high density of vibrational modes at the ZO band range plays a strategic role providing them the unique property of having larger heat capacities than the bulk material at comparable temperatures to room temperature and higher.

This is a non-intuitive result that, to the best of our knowledge, is the first reported system having such property. As a reference, all reported metal nanoparticles thermodynamics results show higher heat capacities than their bulk counterparts only at very low temperatures (~3->40 K).

These new results and discussion were included in an additional section of the revised manuscript (section “Thermodynamic implications”).

Figure R4. Top panels: Heat capacity (red curves) and VDOS (blue curves) for the C720 fullerene (left) and graphene (right). Bottom panel: Difference in the heat capacity (red curve) between the C720 fullerene and graphene. The blue curve corresponds to the difference of the integrals of the VDOS, calculated as a function of frequency. This difference is responsible for a non-zero value of the red curve, corresponding to the difference between the heat capacities of the two systems.

Furthermore, they speculate that in the limit of graphene as a fullerene of infinite radius with very diluted pentagons, the absent modes would emerge with the pentagons having zero-amplitude displacements. These speculations are inconsistent with the fact that an isolated pentagon in a graphene sheet is a disclination, i.e., it induces topological conical curvatures, and even in a sheet with null net curvature (with the same number of isolated disclinations and anti-disclinations) local curvatures would be associated with the isolated pentagons. The scenario of an isolated pentagon having zero-amplitude in the said phonon modes is very unlikely.

Moreover, it seems to me that all that it takes is for an odd number of vertices of the pentagons to have zero-amplitude for the mode to be present in a graphene sheet with isolated diluted pentagons, at least from the point of view of the frustration of the pattern of displacements of the modes in question.

We thank Reviewer 3 for this thorough analysis and accurate description of the phenomenon in a more realistic scenario. The argument we presented in our manuscript was based on a hypothetical scenario in which we considered an ideal physical system with the sole purpose to analyse the limiting case of a fullerene with a very large radius. In order to avoid confusion, we have rephrased our

statement in the main text. Now, instead of analysing the ideal case, we base our discussion on the results presented in Table S1, which support our argument that as the size of the fullerene increases the pentagonal relative displacements decrease.

REVIEWERS' COMMENTS:

Reviewer #1 (Remarks to the Author):

I think the authors have responded all my comments and the work is improved a lot. I recommend publication of this manuscript.

Reviewer #2 (Remarks to the Author):

The authors have revised their manuscript to account for the reviewer comments. The additions further improved the manuscript, in particular the comparison with carbon nanotubes and graphene flakes that have pentagonal defects and the discussion about the influence on the temperature dependent heat capacity as example for the thermodynamic implications. I recommend publication in Communications Chemistry.

Concerning the answer on my comment if the difference in the VDOS can be attributed to finite size effects rather than to the presence of pentagons, the authors may have misunderstood my point. It is evident that figure 2 illustrates the finite size effects, but the question was about the convergence (- i.e. why still a difference at C720). The new calculations for the nanotubes and graphene flakes, however, answer the question. The presented evidence now convincingly demonstrates that the difference in Figure 3 is due to the pentagons and not to remaining finite size effects in C720.

Reviewer #3 (Remarks to the Author):

Given the extended scope of the revised manuscript, and the effort to address all comments by the three Referees, I recommend publication of the work in Communications Chemistry.

I would like to bring to the authors attention two works that, because they are directly related to their manuscript, I would suggest that they be referenced:

1) Gap opening in topological-defect lattices in graphene

J da Silva-Araújo, H Chacham, RW Nunes

Physical Review B 81 (19), 193405

In this work graphene sheets with isolated pentagons are discussed.

2) Nature of localized phonon modes of tilt grain boundaries in graphene WA Diery, EA Moujaes, RW Nunes

Carbon 140, 250-258

Phonon modes of polycrystalline graphene models with pentagon-heptagon grain boundaries.

Response to reviewers' comments and suggestions

We thank all the reviewers for their time and in general for their insightful comments and suggestion that certainly improved the quality and clarity of the manuscript.

REVIEWERS' COMMENTS:

Reviewer #1 (Remarks to the Author):

I think the authors have responded all my comments and the work is improved a lot. I recommend publication of this manuscript.

We thank the reviewer for the positive evaluation and previous helpful comments.

Reviewer #2 (Remarks to the Author):

The authors have revised their manuscript to account for the reviewer comments. The additions further improved the manuscript, in particular the comparison with carbon nanotubes and graphene flakes that have pentagonal defects and the discussion about the influence on the temperature dependent heat capacity as example for the thermodynamic implications. I recommend publication in Communications Chemistry.

Concerning the answer on my comment if the difference in the VDOS can be attributed to finite size effects rather than to the presence of pentagons, the authors may have misunderstood my point. It is evident that figure 2 illustrates the finite size effects, but the question was about the convergence (- i.e. why still a difference at C720). The new calculations for the nanotubes and graphene flakes, however, answer the question. The presented evidence now convincingly demonstrates that the difference in Figure 3 is due to the pentagons and not to remaining finite size effects in C720.

We thank the reviewer for the positive evaluation and the previous helpful suggestions that led to additional insightful results.

Reviewer #3 (Remarks to the Author):

Given the extended scope of the revised manuscript, and the effort to address all comments by the three Referees, I recommend publication of the work in Communications Chemistry.

I would like to bring to the authors attention two works that, because they are directly related to their manuscript, I would suggest that they be referenced:

1) Gap opening in topological-defect lattices in graphene

J da Silva-Araújo, H Chacham, RW Nunes

Physical Review B 81 (19), 193405

In this work graphene sheets with isolated pentagons are discussed.

2) Nature of localized phonon modes of tilt grain boundaries in graphene WA Diery, EA Moujaes, RW Nunes

Carbon 140, 250-258

Phonon modes of polycrystalline graphene models with pentagon-heptagon grain boundaries.

We thank the reviewer for additional references, which after carefully revising them we have decided to cite them in the manuscript given that they support the narrative and arguments of our work.